# Correlation of Body Composition and Nutritional Status with Functional Recovery in Stroke Rehabilitation Patients

**DOI:** 10.3390/nu12071923

**Published:** 2020-06-29

**Authors:** Hiroshi Irisawa, Takashi Mizushima

**Affiliations:** 1Department of Rehabilitation Medicine, Dokkyo Medical University, 880, Kitakobayashi, Mibu, Shimotsuga, Tochigi 3210293, Japan; mizusima@dokkyomed.ac.jp; 2Department of Rehabilitation Medicine, Setagaya Memorial Hospital, 2-30-10, Noge, Setagaya, Tokyo 1580092, Japan

**Keywords:** stroke, rehabilitation, body composition, nutrition

## Abstract

Previous studies have suggested that the nutritional status after stroke is independently associated with long-term outcomes and that sarcopenia delays poststroke rehabilitation and worsens the prognosis. However, many stroke patients have a deteriorated nutritional status and a decreased muscle mass in the acute phase. This prospective study included 179 patients who were admitted to the stroke rehabilitation unit. We performed bioelectrical impedance analysis and determined the Geriatric Nutritional Risk Index (GNRI) to assess muscle mass and the nutritional status on admission. Furthermore, we analyzed the activities of daily living using the Functional Independence Measure (FIM) at the time of admission and four weeks later. Furthermore, we evaluated the change in motor FIM items and examined the relationship with the data. Multiple regression analysis revealed that a high muscle rate (skeletal muscle mass/body weight) (odds ratio OR = 2.43), high phase angle (OR = 3.32), and high GNRI (OR = 2.57) were significantly associated with motor FIM items at four weeks in male and female patients. Muscle mass maintenance through nutritional management and early rehabilitation in the acute period of stroke is essential for functional recovery in stroke patients.

## 1. Introduction

Among stroke patients, patients in severe condition undergo intensive rehabilitation in the stroke rehabilitation unit after acute treatment. On the other hand, disability-adjusted life years related to disability after stroke is increasing [1], and stroke continues to be a major cause of disability in the elderly. Thus, it is important to intervene in the damage caused by stroke and the resulting decline in physical function, and various studies are currently being conducted.

It is known that lower limb muscle strength is related to the walking ability and activities of daily living (ADL) among stroke patients [2,3]. Skeletal muscle mass is related to muscle strength, and the diameter and tension of extracted human muscle fibers are highly correlated [4]. In stroke patients, there have been many studies showing the relationship between lower limb skeletal muscle mass and strength [5,6,7,8,9]. On the other hand, in recent years, the correlation between muscle mass and muscle strength has been found to be only moderate, and it has become clear that muscle weakness cannot be partially explained only by muscle mass reduction [10,11,12]. The reason is that intramuscular extracellular fat and extracellular fluid are present in the muscle cell gaps of skeletal muscles. Ryan et al. reported that the intramuscular fat mass on the paralyzed side increased by about 25% compared with that on the non-paralyzed side in measuring intramuscular fat mass in stroke patients using computed tomography (CT) [8]. Skeletal muscle mass in stroke patients should be considered along with this intramuscular fat.

For these reasons, there is increased need for body composition methods with greater sensitivity and precision. Different methods have been developed to determine the body composition with different physical principles using different models and assumptions. Tomographic imaging techniques such as computed tomography (CT) and magnetic resonance imaging (MRI) are available, and involve in vivo measurements of different fat depots and fat infiltration in organs and are considered to be the gold standard for body composition analysis. Each of the methods has advantages and disadvantages. Bioelectrical impedance analysis (BIA) is commonly used for assessing total body composition in clinical and community wellness settings due to its rapid, non-invasive methods that are relatively inexpensive and widely accessible.

In addition, a poor nutritional status in poststroke patients is known to be an inhibitor of the recovery of physical function [13]. Malnutrition causes a decrease in body cell mass (BCM), and much of the BCM is occupied by skeletal muscle. Therefore, nutritional assessment and nutritional intervention are important in patients undergoing stroke rehabilitation, as malnutrition results in decreased physical function through skeletal muscle loss. To assess nutritional status, the geriatric nutritional risk index (GNRI) was reported to be a very simple and objective method based on body weight, height, and serum albumin levels to assess the nutritional status in a number of pathological conditions.

As described above, the evaluation of skeletal muscle mass and nutritional status is important in stroke rehabilitation, but the effects on ADL recovery have not been clarified. The purpose of this study was to clarify the effects of body composition measured by BIA and nutritional status measured by GNRI on ADL recovery during stroke rehabilitation.

## 2. Materials and Methods

This prospective study was conducted at two stroke rehabilitation units in Japan between January 2017 and June 2018. All subjects gave their informed consent for inclusion before they participated in the study. The study was conducted in accordance with the Declaration of Helsinki, and the protocol was approved by the Ethics Committee of Setagaya Memorial Hospital (H30-001). The study included 210 consecutive patients with stroke. Patients with a pacemaker, a high ADL score (motor FIM items > 80), severe cognitive impairment, sever dysphasia, and early discharge were excluded from the study (Figure 1). Eventually, 179 patients (90 female and 89 male patients; mean age 79.7 years) were included.

### 2.1. Bioelectrical Impedance Analysis

BIA was performed using the InBody S-10 analyzer (InBody Japan, Tokyo, Japan), which applies a 200-μA current at frequencies of 5, 50, and 250 kHz after 10 min of rest at an ambient temperature. In all patients, the BIA was performed immediately after admission to the rehabilitation unit. For three hours before the measurements, the patients did not consume any liquids or solids. The same operator performed the analysis in all patients. For the analysis, four electrodes were placed on both the hands and both feet of each patient in the supine position. The electrodes were placed on hairless skin of the hands and feet. The body weight of the patients was measured during hospitalization using a folding stretcher, and its empty weight was subtracted from the total weight. In the BIA, total body water composition, total body fat, skeletal muscle mass, and the phase angle were measured. The phase angle for the whole body at 50 kHz was calculated from the impedance values. In order to standardize the values, we determined the body water composition percentage, body fat percentage, and body muscle percentage by dividing the total body water composition, total body fat, and skeletal muscle mass by weight. In addition, the blood albumin level was measured on admission.

### 2.2. Nutritional Status

Based on the report of Wakabayashi [14], the meal during hospitalization was 1500–2000 kcal for each patient (protein 1.5 g/kg/day). To evaluate the nutritional status, we assessed the GNRI on admission to the stroke rehabilitation unit. The GNRI has been described by Bouillanne et al. [15]. The GNRI is universally adopted to evaluate a patient’s nutritional status. It is an effective and simple risk index that evaluates a patient’s nutritional risk and has been proven as a predictive index for prognosis in the elderly, dialysis, and cardiovascular patients, and health care. All patients’ nutritional statuses were evaluated according to the GNRI formula as follows: GNRI = (1.489 × albumin [g/L]) + (41.7 × [weight/WLo]), where WLo stands for ideal weight and was calculated using the Lorentz equation (for men: H − 100 – [(H – 150)/4]; for women: H − 100 − [(H – 150)/2.5]; H: height).

### 2.3. Functional Measurements

To evaluate the ADL status, we assessed the Functional Independence Measure (FIM) motor scores of the patients. The FIM contains 13 items on motor tasks, and all are rated on a 7-point ordinal scale, with higher scores indicating more independence [16]. The scale has been used mainly in neurological rehabilitation (including patients with stroke and brain injury) as well as geriatric rehabilitation [17]. The FIM was scored by members of the multidisciplinary rehabilitation team on the day of admission to the stroke rehabilitation unit and four weeks later. The amount of change in the motor FIM score over four weeks was determined, and was used as the value for functional recovery.

### 2.4. Stroke Rehabilitation Unit in Japan

In Japan, it is recommended by the National Health Insurance to shift a patient from the emergency unit to the rehabilitation unit as soon as possible after the end of acute treatment for conditions such as stroke. In this study, we focused on a rehabilitation unit specializing in stroke. In the stroke rehabilitation unit, stroke patients are provided with less than 180 min a day (seven days a week) of rehabilitation according to their symptoms, under the direction of a rehabilitation doctor.

### 2.5. Statistics

Continuous variables were expressed as mean ± standard deviation (SD). Independent *t*-tests were used to detect possible differences between male and female patients. The relationships between sex, advanced age (>80 years), no malnutrition (GNRI > 92), high body fat percentage (male > 25%, female > 30%), high body muscle percentage (male > 30%, female > 25%), high body water composition percentage (>60%), high phase angle (male > 3.5°, female > 3.0°), and functional recovery were estimated using odds ratios and 95% confidence intervals obtained from multivariate logistic regression models. Since there are gender differences in body muscle mass, body fat mass, and phase angle, the analysis was performed on a gender basis [18]. We set the cut-out values of body fat percentage and body muscle percentage according to the criteria for the elderly Japanese population [19]. All statistical analyses were performed using IBM SPSS Statistics ver. 25 (IBM Corp., Armonk, NY, USA).

## 3. Results

### 3.1. Descriptive Characteristics

The descriptive and functional characteristics of the study participants are presented in Table 1.

All of the participants were Japanese (Asian). The mean admission time to the stroke rehabilitation unit after onset was 27.6 days. The participants received the stroke rehabilitation program for about 160 min per day in the stroke rehabilitation unit. Male participants were significantly taller (*p* < 0.05) and heavier (*p* < 0.05) than female participants; however, there were no significant age or body mass index (BMI) differences between the male and female participants. The motor FIM items was significantly increased in both male and female in four weeks (*p* < 0.05, respectively). Additionally, there were no significant differences in nutritional status and functional ability and recovery between male and female participants. In the BIA, there were significant differences in body fat percentage, body muscle percentage, and the phase angle between male and female participants (Table 2).

Male participants had a higher body muscle percentage and phase angle than female participants. On the other hand, female participants had a higher body fat percentage than male participants.

### 3.2. The Univariate Analyses

We then investigated which covariates were associated with functional recovery. In the univariate analysis, no malnutrition, a high body muscle percentage, and a high phase angle were associated with functional recovery (Table 3).

## 4. Discussion

This is the first study to clarify the relationships between BIA, nutritional status, and functional recovery in stroke patients. The nutritional status, muscle mass, and phase angle at the start of intensive stroke rehabilitation exerted considerable influences on functional recovery.

Stroke patients experience hemiplegia due to pyramidal tract disorders. It has been reported that the muscle mass of chronic stroke patients is significantly lower on the paralyzed side (both the upper and lower limbs) than on the non-paralyzed side [20]. This decrease in muscle mass was defined as stroke-induced sarcopenia by Scherbakov et al., and the authors reported that this phenomenon is caused by a combination of disuse muscle atrophy, spasticity, inflammation, denervation, and re-innervation, and nutritional deficiencies [21]. In addition, it has been reported that more rehabilitation from the early phase of stroke improves ADL [22,23,24].

As a method for measuring skeletal muscle mass, assessment of the limb circumference was used as the simplest approach. Local muscle mass measurements involved CT, magnetic resonance imaging, and ultrasound. Dual-energy x-ray absorption and BIA were used to measure muscle mass throughout the body.

In the BIA method, a weak current is passed through the body, and its electrical impedance is used to indirectly determine the amount of water, body fat, and muscle mass. Although it is minimally invasive and simple, it has the problems of being affected by body water issues such as dehydration and edema, and was affected by changes in conductivity due to body temperature [25].

There were sex differences in the body fat percentage, body muscle percentage, and the phase angle. These findings were consistent with previous results [19,26]. Therefore, the analysis had to be performed separately for male and female participants. Our study revealed that both male and female participants showed faster functional recovery when body muscle percentage was high and the phase angle was high. Body fat percentage and body water composition percentage did not affect functional recovery.

The phase angle is the most frequently applied BIA parameter in the clinical setting. It reflects both the quantity and quality of soft tissue and is currently regarded as a composite measure of tissue resistance and reactance [27]. The phase angle increased according to the structural completeness of the cell membrane and the improvement of cellular function and decreased when the plasma luminal structural damage of the cell is present. In a pure cell membrane mass, the phase angle is 90 degrees, whereas that in the pure electrolyte water is 0 degrees. In healthy subjects, the phase angle typically ranges from 8 degrees to 15 degrees [27]. A previous study also showed that male and younger subjects had a higher phase angle [27]. The decrease in the phase angle with increasing age might reflect cell function and general health conditions in addition to body composition [28]. The phase angle in our study participants was lower than that in the previous study. It is speculated that cell function and general health conditions are reduced in stroke patients compared with healthy elderly individuals.

Phase angle analysis is also known to be useful for evaluating the nutritional status. There was a significant association between a low phase angle and the nutritional risk, hospitalization length, and mortality rate. In patients with pancreatic cancer, the phase angle has been shown to be a better predictor of mortality than the traditional nutrition index [29] or weight loss, because the change in the phase angle that occurs before cachexia appears earlier than weight loss [29]. A low phase angle indicates the presence of malnutrition and reflects the development of edema due to acute inflammation or hypoalbuminemia [30].

In this study, we used the GNRI for the assessment of the nutritional status. The GNRI is an indicator of the nutritional status as reported by Bouillanne et al. [15], and a value of 98 or higher is considered to indicate a good nutritional status. In this study, the mean GNRI of the participants was lower than 98, and both male and female participants were considered to have malnutrition. It has been reported that malnutrition often occurs after stroke and that the nutritional status in the acute phase affects functional recovery [31]. This study evaluated the nutritional status in the stroke rehabilitation unit. As a result, it became clear that the nutritional status also affected functional recovery. Malnutrition has been reported to be caused by starvation, acute illness, and chronic illness [32], and patients undergoing stroke rehabilitation may have any of these conditions. In other words, the patients are elderly stroke patients, often suffering from nutritional deficiencies after acute treatment and chronic diseases such as chronic heart failure. These can all cause malnutrition. It has been reported that elderly people require more protein than younger adults, that the reduction in protein associated with malnutrition may reduce muscle mass [33], and that the combination of protein intake and exercise therapy can increase muscle mass in elderly people [34]. In the present study, we reported that stroke patients with a low GNRI had poor functional recovery after rehabilitation. Additionally, the results suggested that the evaluation of muscle mass related to functional recovery and nutrition intervention is necessary. However, it is possible that stroke patients with cachexia had a low GNRI. The effect of rehabilitation on cachexia is not known [35]. Mobility in stroke patients with cachexia having a low GNRI may have been poor because it is difficult to obtain positive effects of rehabilitation in these patients. However, we also report that there was no relationship between body water composition and functional recovery. It may be necessary to examine the relationship between cachexia and body water composition.

There are several limitations in this study. First, we did not follow changes in body composition and nutritional status. It is necessary to examine how body composition and nutritional state change as function recovers, and we wish to assess these changes in the future. Second, we only evaluated muscle mass and not muscle strength. It has recently been reported that muscle mass does not always reflect muscle strength [10,11,12], and thus, further study is necessary.

## 5. Conclusions

From this study, it is clear that the nutritional status and muscle mass at the start of intensive stroke rehabilitation exerted considerable influence on functional recovery. This indicates that nutritional management and rehabilitation for maintenance of muscle quality are important in the acute phase of stroke. Although the phase angle is used as a prognostic factor for various diseases, it is useful for predicting the functional prognosis of stroke patients.

## Figures and Tables

**Figure 1 nutrients-12-01923-f001:**
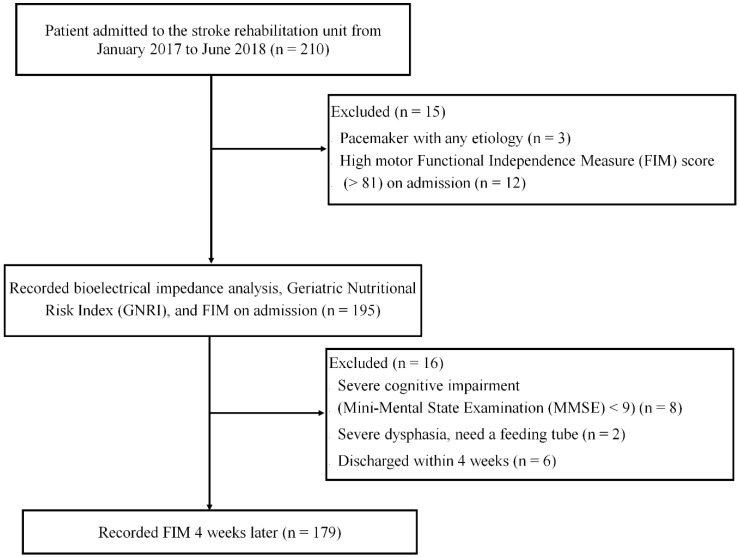
Flowchart outlining the inclusion and exclusion criteria and study design. The study included 210 consecutive patients with stroke. Patients with a pacemaker, a high ADL score, severe cognitive impairment, severe dysphasia, and early discharge were excluded from the study.

**Table 1 nutrients-12-01923-t001:** Characteristics of the study patients.

Characteristic	Mean	SD
Number of patients	179	
Age (years)	79.7	11.5
Sex (F/M)	90/89	
Mini-Mental State Examination	20.2	8.0
Days after stroke	27.6	8.7
Duration of rehabilitation program (min/day)	159.8	21.6
Motor Functional Independence Measure items on admission	39.0	19.9
Motor Functional Independence Measure items at 4 weeks	53.6	26.8

**Table 2 nutrients-12-01923-t002:** Characteristics of patients according to sex.

	Men (*n* = 89)		Women (*n* = 90)	
	Mean	SD	Mean	SD
Age (years)	78.6	13.3	80.7	8.3
Height (cm)	158.1	13.5	153.2 *	7.1
Weight (kg)	52.4	15.8	45.7 *	10.3
BMI (kg/m^2^)	20.0	3.76	19.4	3.9
Albumin (g/dL)	3.8	0.4	3.7	0.4
GNRI	97.3	10.4	93.5	11.3
motor FIM items on admission	39.5	18.5	38.6	21.5
motor FIM items on 4 weeks	55.7	27.6	52.8	25.4
BIA data				
Body fat percentage (total body fat/body weight) (%)	24.2	8.5	28.8 *	10.9
Body muscle percentage (skeletal muscle mass/body weight) (%)	40.0	4.8	35.8 *	5.7
Body water percentage (total body water/body weight) (%)	55.7	6.0	52.3	8.0
Phase angle (%)	4.2	1.1	3.3 *	0.9

* means *p* < 0.05.

**Table 3 nutrients-12-01923-t003:** Associations between functional recovery and clinical covariates.

Variables	Odds Ratios	95% CI	*p* Value
Male gender	2.11	0.86–5.21	0.10
Advanced age (>80 years old)	1.20	0.48–2.98	0.69
No malnutrition (GNRI > 92)	2.57	1.13–5.85	0.02
High body fat percentage (male > 25%, female > 30%)	1.81	0.70–4.70	0.69
High body muscle percentage (male > 30%, female > 25%)	2.42	1.05–5.59	0.03
High body water composition percentage (>60%)	2.84	0.50–16.23	0.24
High phase angle (>3.0 degree)	3.23	1.37–7.65	0.01

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
