# Peer review of "Correlation of Body Composition and Nutritional Status with Functional Recovery in Stroke Rehabilitation Patients"

_nutrients, 2020, doi:10.3390/nu12071923_

Round 1

Reviewer 1 Report

The present study aims to investigate an influence of the body composition and nutritional status on functional recovery in patients with stroke undergoing the post-stroke rehabilitation program. The study is well designed, clinically relevant and provides additional evidence suggesting that muscle mass and nutritional status contribute to the functional recovery after stroke.

Minor Comments

  1. I assume that the study was prospectively conducted according to the study protocol at the rehabilitation unit, and merely the data analyzing was performed retrospectively? If so, the study is a prospective and this should be so indicated in the Methods.
  2. The univariate analyses could be placed in the separate section (e.g. 3. Results 3.2. Logistic regression analyses)
  3. How you explain that the phase angle was significantly lower in women although there were no significant differences in age, albumin, nutritional status and functional dependence between men and women? Was an improvement of functional recovery according to FIM significant during rehabilitation?
  4. The Discussion section needs to be shortened and better structured addressing the results of the study more precisely. Please rewrite you conclusions according to the results.

Author Response

Thank you for reviewing of our paper. We have answered each of your points below.

  1. We appreciate the reviewer's comment on this point. In accordance with the reviewer's comment, we have changed this to “prospective study” on line 15 and 57.
  2. We strongly appreciate the reviewer's comment on this point. Accordingly, we have added “2 The univariate analyses” to this section.
  3. The reviewer's comment is correct. To clarify, we have added the following text to the Results (lines 139-140): The motor FIM items was significantly increased in both male and female in 4 weeks (p < 0.05, respectively). According to reference [27], the phase angle is generally considered to be higher in male, which is consistent with the results in this study. We have added the following text to the Discussions (line 190): A previous study also showed male and younger subjects having higher phase angle.
  4. In accordance with the reviewer's request, we have shortened the Discussions. We have had the Conclusion rewritten.

Reviewer 2 Report

The paper describes a connection among  body composition and nutritional status in store rehabilitation patients. The experiment is well planned and controlled. The number of the participants is sufficient to draw clear conclusions.

The statistical analysis seems to be appropriate, but occasionally very large deviations appear. Was a power calculation performed to estimate the size of the group?

My remarks or questions:

Is there any precise information about the diet of the participants (meat/week, type of meat for example)?

The text and Table 1 show that there were 90 women and 89 men. In Table 2, 79 men and 80 women were analysed. Please explain why?

Is the percentage of people with osteoporosis among the respondents known?

Minor changes:

  1. ADL (line 53) or ADLs (line 36) - please standardize
  2. Line 61 - Please specify the number of points of ADL scale
  3. Figure 1 – Please expand the abbreviation when it first appears (FIM and GNRI)

Author Response

Thank you for reviewing of our paper. We have answered each of your points below.

Questions

  1. We agree that this point requires clarification, and have added the following text to the Methods (lines 89-90): Based on the report of Wakabayashi [14], the meal during hospitalization was 1500-2000 kcal for each patient (protein 1.5 g/kg/day).
  2. The error on Table 2 has been corrected in accordance with the reviewer's comment.
  3. We do not evaluate the proportion of osteoporosis in subjects.

Minor changes

  1. Accordingly, we have changed this to ADL.
  2. Accordingly, we have added (Motor FIM items > 80) to this section.
  3. As requested, we have expanded the abbreviation in Figure 1.